# Exo Supportive Devices: Summary of Technical Aspects

**DOI:** 10.3390/bioengineering10111328

**Published:** 2023-11-17

**Authors:** António Diogo André, Pedro Martins

**Affiliations:** 1Associated Laboratory of Energy, Transports and Aeronautics (LAETA), Biomechanic and Health Unity (UBS), Institute of Science and Innovation in Mechanical and Industrial Engineering (INEGI), 4200-465 Porto, Portugal; a.diogo.andre@icloud.com; 2Faculty of Engineering, University of Porto (FEUP), 4200-465 Porto, Portugal; 3Aragon Institute for Engineering Research (i3A), Universidad de Zaragoza, 50018 Zaragoza, Spain

**Keywords:** external device, biomechanical design, structural materials, actuation, energy sources, control system

## Abstract

Human societies have been trying to mitigate the suffering of individuals with physical impairments, with a special effort in the last century. In the 1950s, a new concept arose, finding similarities between animal exoskeletons, and with the goal of medically aiding human movement (for rehabilitation applications). There have been several studies on using exosuits with this purpose in mind. So, the current review offers a critical perspective and a detailed analysis of the steps and key decisions involved in the conception of an exoskeleton. Choices such as design aspects, base materials (structure), actuators (force and motion), energy sources (actuation), and control systems will be discussed, pointing out their advantages and disadvantages. Moreover, examples of exosuits (full-body, upper-body, and lower-body devices) will be presented and described, including their use cases and outcomes. The future of exoskeletons as possible assisted movement solutions will be discussed—pointing to the best options for rehabilitation.

## 1. Introduction

Physical impairment limitations are still a common occurrence in today’s society, despite the advancements in technology and science, and the implementation of new legislation in most countries, defining new rules for facilities. These physical impairments have many causes [1], such as the normal aging process and increased average life expectancy [2,3], neurodegenerative diseases [4], and accidents including falls, motor vehicle accidents, or sports-related incidents [5].

### 1.1. Context and Demographics

The average life expectancy of a nation is proportional to its level of development. Countries with higher quality of life (QOL) indices tend to have longer average lifespans than those with lower QOL indices. For example, Norway, Germany, and Hong Kong all have an average lifespan of over 80 years whereas the Central African Republic and Nigeria both have an average lifespan below 54 years. Therefore, countries should focus on improving their overall standard of living if they wish for their population to live longer lives beyond retirement age.

The aging population is becoming a global phenomenon, mostly in developed countries. Statistics show that in the United States of America (USA), the number of people aged over 65 years old will be approximately 55 million in 2030 [6]. This is similar to Europe, where the number of elderly people already represents almost 20% of the total population [7]. Surveys from the United Nations (UN) reveal that by 2050, around 20% of the global population will be over 60 years old [8], and of them, 1.5 billion will be over 65 years old [9]. These provisional numbers more than duplicate the 2015 and 2019 numbers, respectively, [9,10]. As an inherent consequence of an aged population, locomotion disorders became a reality for those people [11]. However, they are not age-exclusive results. Neurological pathologies, characterized by the progressive loss of structure and function of the central nervous system caused by neuron death, are also responsible for them. Without a healthy nervous system, sensory information (audition, vision, smell, tact, and taste) and well as muscle coordination are compromised.

The global prevalence of dementia estimated by the UN points to more than 55 million people worldwide having some kind of dementia and 10 million new cases diagnosed each year [12,13,14]. Parkinson’s disease (6.2 million in 2015 [15]) and Alzheimer’s disease (60–70% of all cases [14]) are the two most common forms, along with amyotrophic lateral sclerosis. As these neurodegenerative disorders progress, they cause a gradual decline in patients’ locomotion abilities, leading to death within two years for many patients [16]. Despite extensive research into treatments for these diseases, there is still no effective treatment available.

Spinal cord (SC) injuries are a major source of locomotion disorders. These changes in muscular strength or bodily function can be either permanent or temporary and affect everyday activities such as walking and lifting a glass of water. These injuries can be caused by external traumas, such as car accidents (39.3%), falls (31.8%), gunshots (13.5%), or during sports (8%) [17]. They may also be caused by tumors (33–79%) [18], which compress the SC. However, depending on the extent of SC damage, it is classified as complete (no messages are conveyed to body parts) and incomplete (some level of message transmission is still possible). Every year about 40 million people suffer from SC injuries, most of them between 20 and 35 years old [19].

In some scenarios, locomotion disorders are multi-factorial, making movement tasks much more challenging. One example is the simultaneous occurrence of tumors and neurodegenerative disorders, since their risk increases with age [20]. All these adversities have a high and long-term impact on the social, economic, and financial spheres, affecting communities and healthcare systems worldwide [3,8].

Psychologically, locomotion problems may be a cause of stress, pain, and depression, since the simplest movements are no longer easily achievable. Moreover, motion disorders also contribute to depressive states of mind, as they negatively impact a person’s social life [6,21,22]. Assuming these bad feelings, mental illness is often found in people who are experiencing physical impairments, affecting not only the person but also family and friends.

Bedridden patients (or immobile patients) tend to develop a condition called “sarcopenia”, which is the deterioration of muscle tissue that leads to immobility [23]. In addition, the risk of comorbidities, such as obesity, coronary heart disease, and diabetes increases [24].

Researchers, engineers, and physicians at universities, research institutes, or companies [25] are working to address the consequences of aging and injuries that affect human movement. Several solutions, such as walkers, wheeled vehicles, and wheelchairs [26] have been available for decades. These assistive technologies are meant to help the affected person regain some independence. However, these simple devices were not designed with rehabilitation in mind therefore, some exoskeletons were developed to fill that void [27].

### 1.2. Concept of an External Device

The concept of an exoskeleton has its roots in the natural world. Some animals, such as arthropods and mollusks, have a hard outer layer called an exoskeleton (distinct from the endoskeleton found inside the body of others), which serves to protect their bodies from the elements and provides a surface for muscle attachment and a barrier from dehydration, besides a sensory interface to the surrounding environment [28].

For humans, exoskeletons, which emerged in the 1950s [25], are systems that can expand or enhance a person’s physical abilities [28]. These mechanical devices are fitted with powerful actuators at human joints, allowing for assisted movement [29]. Originally developed for military use, such as aiding soldiers with carrying heavier loads, running faster, jumping higher, or fighting better [28], exoskeletons are now being used and developed for different purposes, such as for medical applications (e.g., assisting physiotherapy [30]) and for industrial purposes.

Firefighters and other rescue workers have been using exosuits in their daily activities to help them carry heavier loads. Additionally, certain industrial companies have also been equipping their employees with passive external skeletons to help reduce fatigue and increase productivity [31]. Although the range of applications that exoskeletons and exosuits are already used for is wide, they are still being actively developed and improved upon, as shown by the increasing number of publications on the subject. In 2014, the number of papers published on the topic was nearly double that of 1997 [32], demonstrating the remarkable progress being made in all aspects of exoskeleton and exosuit design.

## 2. External Devices in Rehabilitation Context

After a trauma or surgical procedure, continuous passive motion devices are typically used in rehabilitation to reduce edema, bleeding, pain, and inflammation. These devices are the first step in the rehabilitation process. Active assistive movement is also used, which helps the patient perform desired movements with the help of a suit that assists in completing the movement. In cases of neurological rehabilitation, this method is the first choice to stimulate neuroplasticity and reduce common side effects such as muscle weakness. Active resistive motion involves applying an external resistive force against a dynamic or static muscle contraction and is an effective way to increase bone and muscle mass, making it essential for musculoskeletal rehabilitation. Exoskeleton usage can enhance the results of different physiotherapeutic approaches. Still, the final outcome depends on a range of rehabilitation factors, including timing, intensity, repetition, frequency, and task-specific training protocols [33].

Wearing an external device, such as an exoskeleton, can provide numerous advantages in a medical rehabilitation environment [34], not only for the patients but also for clinical centers. These devices can enable patients to perform intensive and repetitive movements with precision, minimizing the physiotherapist’s intervention [35]. This can relieve therapists from fatigue and constant attention requirements. Additionally, this kind of technology can enable the rehabilitation of patients at their homes via video conference. Exoskeletons can also be used to evaluate recovery levels by measuring force levels and movement patterns [8]. This data can be collected from sensors [36] in the device itself and/or from motion capture devices that track motion patterns. This training can help people relearn lost motor functions and perform daily tasks.

The drawbacks of existing solutions should be the object of careful consideration, taking into account the person and their particular circumstances. For example, some solutions may not be energy efficient, leading to high energy consumption [26], while others may make it difficult for the user to interact effectively with their surroundings [37].

When rehabilitating a patient using an exoskeleton, the need for a large empty room must be taken into account. Moreover, since a one-size exoskeleton cannot accommodate all users due to differences in body proportions, the creation of an adjustable device that can fit all sizes poses a great challenge due to its complexity. Thus, a disproportional device regarding the body may have a negative psychological impact on the user, leading to some reluctance to make use of it [38].

Despite all the challenges, researchers have already developed reliable solutions to rehabilitate or enhance various parts of the body, such as ankles [39,40,41], hands [33,42], shoulders [43], lower limbs [44], upper limbs [45], arms [46], and back [47,48].

### 2.1. Mechanical Design

Design is an imperative aspect to consider in exoskeleton development, as every detail affects the user experience, with the final appearance being the first overall impression. During conception, several design considerations come into play during all stages of the project, from selecting structural materials to selecting control systems, with particular attention paid to key components such as batteries. An intelligent arrangement of actuators and energy sources (e.g., batteries) brings benefits beyond just aesthetics; it can improve weight distribution [27,49] and in some cases even reduce power consumption [50], which is directly linked to the choice of power source. Most importantly, a good design can make a positive first impression on the users, providing them with a sense of comfort, ergonomics, confidence, and convenience.

In addition to visual appearance, it is crucial to consider the technical aspects when designing the final solution. The movement’s kinematic and dynamic degrees of freedom (DOF) found in the human body [51] based on anthropometry should be present, as a concept, throughout the projects. The range of motion, joint torque requirements, joint rotational velocity, and joint angular bandwidth [52] must also be factored in. The developed device aims to aid and follow human movement without constraint or interference with the natural freedom of movement [53].

Based on the above-stated principles and keeping in mind the intended purpose, the wearable device should enable fundamental body movements, as described in [54]. These movements involve pairs of opposite gestures, such as flexion and extension of hand movements (depicted in Figure 1a), or abduction and adduction of the fingers (illustrated in Figure 1b). Additionally, rotation (medial or lateral), as shown by elbow rotation (Figure 1c), is another basic movement. By combining these basic movements, a person can perform complex movements like writing.

Creating and implementing practical solutions can be a challenging task due to the inherent complexity of the principles involved and their combination. A specific example of this complexity can be observed in the development of complete limb external skeletons. These particular devices are capable of an infinite number of combined movements, as they rely on seven distinct DOF [50] positioned along the limbs. These DOF are vital for daily activities [55], with lower limbs having three DOF at the hip, one at the knee, and three at the ankle, while upper limbs possess three at the shoulder (abduction-adduction, flexion-extension, and internal-external rotation), one at the elbow, one at the forearm, and two at the wrist.

Providing the necessary DOF for full-body applications becomes challenging with traditional exoskeletons, which often consist of rigid materials assembled in a series of fixed links. Their non-flexible characteristics can lead to problems of hyperstaticity [56] and can result in increased device complexity, which further complicates the design process. As an alternative, soft structures composed of mechanisms without rigid components, featuring elastic or elastomeric materials with softer (more flexible) mechanical properties, have emerged. As demonstrated by successful lightweight and flexible designs [57] and greater adaptability to both movement and the human body [58], they offer a promising alternative to their traditional rigid counterparts.

The primary function of the wearable device is not just to track human movement, but also to provide assistance by generating the necessary force or moment to hold the joints (e.g., elbow) in certain positions during daily activities or rehabilitation. Moreover, as per [59] guidelines, the device should be capable of generating the appropriate amount of auxiliary force or momentum to perform those daily tasks. However, it may be impossible to devise a solution that combines all the necessary DOF with adequate motion generation, as illustrated by the challenge of creating a wearable finger device. This limb is essential for performing basic daily tasks, such as typing or writing, and any solution must be practical and effective in addressing these needs. For example, opening a jar with a finger wearable device would require up to 120 N of force and 3 Nm of torque on the metacarpophalangeal (MCP) joint [60], (Figure 2), without neglecting other considerations such as overall aesthetic and having four DOF [61,62].

The project must prioritize security measures, as an equal fundamental design factor. Following these safety standards avoid accidents and ensure the user’s protection in unforeseen events, like power loss or current leaks. The probability of accidents using exoskeletons is real and remains a significant concern, since estimations suggest around 4 out of every 100 users may encounter issues [63]. The overall solution’s appearance, functionality, safety, and ease of use are determined by the final design concept, which cannot disconnect design choices from fundamental design variables and options. Consequently, the final solution must represent a balance between design options, structural materials, actuators, energy sources, and control systems to achieve the best overall solution.

### 2.2. Structural Materials

While the terms exoskeleton and exosuit have often been used interchangeably, some argue that exosuit is a more accurate description for these devices even though the general public is more acquainted with the term exoskeleton. Despite their similarities, those terms are not synonyms when the context involves structural materials. In reality, these words represent two distinct approaches to solving the same problem. Exoskeletons are typically constructed of rigid and metallic components [64,65], while exosuits are designed using soft and flexible materials [66,67]. Although they are classified differently, both solutions should be investigated together as they offer complementary features [68]. Regardless of the type of material used, it is essential that any solution designed for use in rehabilitation settings meets certain critical requirements that ensure safety, as mentioned in a study by Xiloyannis [49]. In particular, mechanical properties assume great importance since patients undergoing rehabilitation are often susceptible to minor accidents, such as small falls, and the material should be able to withstand and resist fatigue, as pointed out by Bogue [69]. These characteristics are vital for ensuring that the device has a long lifespan, even when deployed on a higher number of patients during rehabilitation. Additionally, the material should offer a warm and comfortable sensation to the wearer.

#### Rigid vs. Soft Materials

When it comes to a rigid approach, materials like stainless steel [69], aluminum [70], and titanium [65] are widely used. The final solution can involve one or multiple materials for example, with frames made from aluminum and joints made from stainless steel or titanium. This multi-material approach can offer several benefits, such as reducing weight and increasing mechanical strength at critical joints. In fact, using multiple materials is becoming popular in engineering because it provides a better balance between performance, cost, and durability.

Compared to exosuits, more rigid solutions offer some advantages but bring some disadvantages. Exoskeletons offer increased mechanical strength, making them an ideal solution when high levels of torque and strength are required. In fact, these devices can withstand up to 1 GPa of tension before experiencing plastic deformation and can endure up to 50% of strain before reaching a breakdown point, as shown in Figure 3 [71]. Such impressive performance metrics highlight the potential benefits of using exoskeletons in various settings.

However, the materials used are typically heavier, as shown in Figure 4a, which can limit their portability and cause discomfort for the user [72]. Additionally, achieving perfect alignment between the device and the user’s joints can be a challenge, resulting in larger inertial loads that can lead to abnormal motion patterns [73]. Other common problems associated with rigid solutions include reduced usability and poor aesthetics, as noted by several authors [74,75]. Despite these drawbacks, rigid solutions remain popular in many applications due to their mechanical reliability and stability.

While exosuits and exoskeletons share some common characteristics, such as safety features (when applied to these devices) and price range (as depicted in Figure 4b), there are clear differences in their design and construction. Exosuits typically have symmetric properties not being susceptible to misalignment, largely due to the materials used in their production. These materials primarily consist of polymeric or composite materials, including elastomers such as liquid crystal, dielectric, and acrylic elastomers [66,68], shape memory polymers (SMPs) such as those based on epoxy and polycaprolactone materials [66], electroactive polymers (EAPs) like polyvinylidene difluoride (PVDF) [76,77], and conducting polymers such as polypyrrole [78,79]. Their use in exosuits allows for greater flexibility and symmetry compared to their rigid exoskeleton counterparts.

Composite materials can be comprised of metallic and polymeric substances combined with carbon fibers [57,69]. In some other cases, a solution made from chloroprene and polyurethane (PU) may also be used [57]. Additionally, textiles may also be utilized for certain applications [73,79]. By combining these materials, it becomes possible to create lightweight and durable devices that can provide users with a wide range of benefits.

In general, these materials enable movement smoothness [80], comfort, portability, flexibility, lightweight (low density) [68,73], adaptation to bioorganisms [66] and even the ability to emulate biological muscles [77]. Some of these materials can exceed their structural role and be used as actuators [77] since they are prone to deformations with associated large volume changes in response to external stimuli [66].

However, this approach presents some technical disadvantages. The amount of power that such actuators can transmit and their response in velocity, are highly diminished when compared to rigid solutions due to their (softer) mechanical properties [68]. Therefore, they are especially indicated for low assistance levels [68].

While rigid and flexible materials used in exoskeletons and exosuits possess distinctive properties, the most effective solutions typically involve a combination of both types of materials. Table 1 provides some essential information regarding the characteristics of these different materials and how they can complement each other. By leveraging the unique advantages of both rigid and flexible materials, it becomes possible to create devices that are both durable and comfortable, allowing users to benefit from the qualities of each material type.

### 2.3. Actuators and Energy Sources

Actuators play a critical role in wearable external devices, facilitating human movement by powering them, and in this way, enabling a better interaction with the surrounding environment. In a medical context, they can be particularly valuable for helping patients undergoing rehabilitation by providing controlled motion patterns. As such, actuators are an indispensable component of many modern wearable devices, and their effectiveness can have a significant impact on user outcomes.

Actuators can be classified as either powered or unpowered, resulting in the creation of either active or passive external devices [81], respectively. Powered alternatives may be noisier and are generally costlier due to the need for additional components, as well as requiring users to carry bulky energy-supply systems [82]. On the other hand, passive devices do not require power units, making them lighter and weighing up to a fourth of their powered counterparts. A good example of this is the ankle exoskeleton developed by Mooney et al. [83] and Collins et al. [84], which aims to reduce the metabolic rate during walking [85].

Mooney et al. [83] achieved a weight of 2 kg in their solution, while Collins et al. [44,84] proposed an unpowered solution that was 1.5 kg lighter and cheaper. The essential difference between these two approaches is the presence of either an autonomous system capable of producing its own energy, thus replacing human metabolic sources [83], or a passive system that makes body locomotion more efficient by reusing some of the energy already produced by the body [84]. Also, actuators can be categorized as either traditional or soft, depending on their constituent materials and energy-supply system type (see Table 2). Each alternative exhibits different advantages and disadvantages depending on their intended use.

#### 2.3.1. Traditional Actuators

Traditional actuators typically are based on rigid systems, allowing them to generate higher forces [49], greater movement precision, and improved dynamic performance [68], and as a result, making them ideal for more complex tasks such as severe mobility disorders. However, it brings some disadvantages, such as leading to higher power consumption [68]. When a power supply is required to input the actuator, the user’s freedom of movement can be limited. Additionally, elderly users may feel uncomfortable with the robotic aspect of the actuators, which can convey a detached and cold sensation and lead to their refusal to use the device.

**Purely mechanical actuators** such as springs [86], are commonly used in unpowered devices (which do not require any external source of energy) and convert the tension force from the actuators into torques at the joints [44]. This mechanical solution can help to reduce the metabolic consumption of energy [84] during walking or running activities [87]. However, the usability of such actuators has limited usefulness in rehabilitation cases, as they only provide passive assistance. For example, during walking, the user must first tense the actuator during flexion movement in order to receive assistance in the extension movement.

**Mechanical servomotor-based actuators** [88] are a simple and direct approach for achieving actuation through electrical stimulation. They provide motion and assistance when connected to the structural material (soft or rigid). However, due to the nature of the input type, they always require an external source of electrical energy, such as (portable) batteries. Plus, they are also rigid and bulky, which can limit the flexibility of the entire system [89].

**Pneumatic-based actuators** are a highly efficient and safe solution in terms of linear and rotational movement control since the actuator’s motion is converted from pressurized air energy [89]. Also, they are particularly suitable for applications that demand repetitive opening and closing tasks, as well as in environments of extreme temperatures or even in industrial applications where other types of actuators are not viable alternatives. As air-compressed-based actuators, this type of solution can convert up to 6 bar of pressure into movement, if necessary. However, to perform all of this and enable movement, connectivity to a rigid control and power system, such as a compressor, is a mandatory aspect requirement [89], which can occasionally lead to pressure drops and noise. Moreover, pneumatic actuators could be produced either considering rigid [90] or soft materials, such as latex or rubber tubes [91], which make them a feasible solution for exoskeletons [90] and exosuits [91].

**Hydraulic actuators** [92,93] share similar advantages and disadvantages when compared to pneumatic actuators. Similarly, they require a hydraulic fluid to output linear, rotary, or even oscillatory movements by the actuator, but as liquids are nearly incompressible, the force produced is considerably higher. The exoskeleton/exosuit movement is thus achieved by converting hydraulic into mechanical energy.

#### 2.3.2. Soft Actuators

Actuation solutions based on soft actuators can be a comfortable alternative when used during the rehabilitation process [119] and unlike the traditional methods, they can be stimulated externally by different inputs. The direct incidence of light, heat, electric or magnetic fields results in mechanical movement performed by the actuators [77,89]. They can be thus defined as mechanical and electrical elements whose output/operation varies under different physical, chemical, and/or biological stimuli. Typically, these soft actuators can be built using a different range of materials (see Figure 5), from particles to polymers, such as EAPs [77] or SMPs [120], papers [89], fluids, shape memory alloys (SMAs) [89], hydrogels, liquid materials [66], 2D materials, carbon-based materials [66] or combinations thereof [89]. Despite the numerous alternatives, not all of these soft actuators are viable for rehabilitation cases. The pertinence of their applicability is based on performance parameters such as stress, strain, Young’s modulus, power, energy, and force density [89].

**Electrically responsive** soft materials are flexible and stretchable materials able to convert external electric inputs into mechanical response outputs. Depending on the type of material, they are classified as dielectric elastomer actuators (DEAs), piezoelectric-based actuators, and electrically conducting polymers (ECPs). DEAs have their input-output conversion based on Coulombic attraction. Two flexible electrodes with a potential difference located on separate ends of a compressible membrane are used to obtain the mechanical response from DEAs [89]. They are highly flexible materials with high energy density, strains, and the ability to emulate the behavior of biological muscles [94]. The performance of these materials depends on their stability, breakdown voltage, and dielectric constant of them [89]. However, they generally require high voltages, usually in the kV range, to perform and leakage currents are often observed when high electric fields are applied, especially when the actuator ages [89]. Adding liquid elastomers to DEAs has been proposed as a solution for these limitations [89]. Examples of dielectric materials found in the literature include acrylic elastomers [95], which are highly deformable and possess high viscoelasticity. However, the actuator’s bandwidth could be limited due to these mechanical properties [89]. Other examples include silicone-based materials [96] and PU-based elastomers [97]. The PU-based elastomers have faster reactions and can be cast into various shapes, but they perform significantly lower strains than the dielectric materials [89]. DEA solutions are constantly being researched and developed to enhance their properties in the actuation field [98].

Piezoelectric-based actuators are capable of producing voltage or electric charge in the presence of mechanical or vibrational forces (direct effect) or deformation when electrically-stimulated (indirect effect) [77,99]. These actuators can operate in room conditions for long periods and have a quick response time, typically in the milliseconds range. Also, they can hold strain under activation, inducing relatively large actuation forces [77]. However, their usability in real-world scenarios can be limited by the large AC voltages required [89]. Common piezoelectric materials for actuation and sensors include PVDF and its copolymers [100,101], graphene [102], and zirconate titanate [103], among many others [104]. ECPs [105] are organic polymeric materials obtained by reduction or oxidation reactions [106]. They can conduct electricity with conductivities up to 10^5^ S/cm, achieved through traditional sources, such as batteries or chemical reactions. Moreover, these electrically responsive material types have been powered using biofuels, such as glucose, which shows their potential as an environmentally friendly source of energy [107]. Polypyrrole, a type of ECP obtained by the oxidative polymerization of pyrrole, is characterized by high mechanical properties and chemical stability [106] and has been shown to emulate human biological muscles due to its similar behavior and low voltage operability [107,108]. These characteristics make ECPs an interesting choice due to their biomimetic and biocompatible nature.

**Magnetic responsive materials** have potential applications as actuators since they are easily controllable through magnetic field direction and magnitude, which can penetrate most materials [89]. This feature makes them a promising solution for use in restricted or enclosed areas [109]. This actuation method is based on incorporating magnetic particles and fillers into different soft compounds such as polymers, gels, papers, or fluids [109]. This results in a magnetization profile with variable magnitude and direction [110]. In the presence of a magnetic field, the particles or fillers align to create deformation, bending, elongation, or contraction [89]. These magnetic-based actuators have a fast response time, with literature reporting speeds of up to 100 Hz [111]. However, there are some disadvantages associated with the magnetic coils used to generate magnetic fields. Their large size, high energy consumption and limited control areas where the magnetic field may not be strong enough are some handicaps to consider [89].

**Thermally responsive materials**, including silicone-based elastomer materials [116], liquid crystal elastomers, and synthetic hydrogels [89], can be activated by a thermal source, such as infrared (IR) radiation, thermal radiation, or Joule heating [113]. For instance, shape-memory materials (SMM) [115] can be deformed by external forces and return to their original “memorized” shape under loading or thermal cycles [89]. These materials include SMAs (typically iron-based or copper-based) [114], which return to their original shape when the temperature exceeds a certain threshold after deformation, and SMP materials (PU and thermoplastic PU) [112]. SMPs are cost-effective, have high elastic deformation, and are easy to manufacture [89]. Furthermore, they can be activated remotely, for instance, through laser incidence, and are often safer than electrical fields for biomedical applications [89]. Some of these light actuators are capable of lifting objects that are up to 200 times heavier than their own weight, to up to 5 mm height [113]. However, such thermally responsive materials tend to have slower response times and are less efficient compared to other types of stimuli-based actuators [89].

**Photo-responsive materials** employ photochromic molecules to capture optical signals and convert them into property modifications [117,118]. They represent an attractive wireless alternative, as they can be controlled in small sizes and consume low energy [89]. However, slow actuation speed and mechanical property degradation remain major limitations [89]. Photochromic molecules, such as spiropyran [117], may be added to various materials, such as gels, polymers, and fluids, to render them photoresponsive [89]. They respond to the light spectrum, visible or near-IR) [89].

### 2.4. Control

The majority of external skeleton or suit devices can be analyzed from two distinct perspectives: mechanical and control system, with the former including structural materials, actuators, and sources of energy, and the latter including sensors that ensure interconnection between the device and the user [121]. The control system’s mission is to predict human intention, interpret signals captured by sensors, and send input to actuators, thereby allowing the skeleton to operate in parallel with the human body [122]. In passive devices that lack powered systems, a control system is unnecessary [123]. Refer to Figure 6 for a depiction of the control solution.

#### 2.4.1. Control System Architectures

The control system of external skeleton or suit devices can be categorized into four main architectures—model-based, hierarchy-based, physical parameters-based, and usage-based, as shown in Figure 7 [121]. While none of these architectures have been used individually due to their complexity or effectiveness, they are frequently combined to achieve the desired control of a specific device [121].

In general, **model-based control systems** can be further classified into two types—dynamic and muscular models [124]. The dynamic model reflects the human motion intent by combining inertial, gravitational, Coriolis, and centrifugal effects to model the human body as a series of rigid links connected by joints (bones) [125,126]. The control system of BLEEX is just an example of a dynamic model-based system [50]. This based-type architecture is even developed through different approaches: mathematical, system identification, and artificial intelligence models. To obtain a mathematical architecture for the external device based on the physical characteristics of the system, the system requires a precise dynamic model [121]. For instances in which a dynamic model cannot be adequately developed through theoretical mathematical models, the system identification model is often utilized [127]. The artificial intelligence method is the most popular approach to identifying the dynamic model due to its efficiency [127].

Muscle-based models have also been utilized in exoskeleton control systems. Unlike dynamic models, these models predict the muscle forces generated by human joints as a function of muscle neural activities and joint kinematics [126,128]. This approach, which can be obtained by using parametric or non-parametric models, takes the electrical signal produced by muscles as input and sends force estimation as output to actuators [121]. The parametric muscle model is commonly implemented using the Hill-based model, which refers to muscle contraction and uses the estimated muscle activation level [129,130,131]. It is comprised of three elements: a contractile element, representing force generated by active muscle fibers, a series element, which models the mechanical response of the muscle, and a parallel element which simulates the passive resistance of muscles to stretch [131]. In addition, the output sent from this type of control model is a function of electromyographic (EMG) neural activity and muscle length [121]. In contrast, non-parametric muscle models do not require knowledge about muscle and joint dynamics but they can be the source of control inefficiencies [132] (ex. finite impulse response model).

Shafer et al. [133] developed an ankle exoskeleton controller that uses a control system based on a neuromuscular model. They make conclusions on the effectiveness of their model in providing a wide range of assistance torque and power. Moreover, Song et al. [134] developed a novel model-based control to predict motion trajectories and amplify the forces produced by the user.

The **hierarchy**-based control system, exemplified in Huang et al. [135] and Dinh et al. [136], utilizes a hierarchical structure to manage inputs and outputs. The controllers are divided into three levels: task level, high level, and low level. The task-level controller, which is the highest level, is responsible for performing the designated tasks [121]. The high-level controller adjusts the force of human-external device interaction based on information received from the task-level controller [121]. Finally, the low-level controller is responsible for controlling the position and/or force performed by the exoskeleton joints, therefore contacting directly to the exosuit [121].

Copaci et al. [137] implemented a hierarchy-based control system in an elbow exoskeleton. Using algorithms to process EMG signals, they were capable of generating position and torque references in SMA actuators used for active rehabilitation therapies.

Control strategies such as those utilized in the ARMin [138], RUPERT IV [139], and LOPES [140] exoskeletons use **physical parameters** as a basis for their implementation. These solutions can be classified as either position, torque/force, or force interaction controllers [121]. The low-level controller in the position control scheme ensures that the exoskeleton joints turn to the desired angle, while the torque/force controller regulates the desired force and/or torque [141], and is also classified as a low-level controller [121].

The interaction force controller, typically functioning as a high-level controller, is responsible for providing appropriate assistance to users during a task [121]. This physical parameter controller takes into consideration the force interaction between the user and the exoskeleton, which is considered in an external device [121]. The impedance controller, which accepts position and produces force, or the admittance controller, which accepts force and yields position, can be used to control this physical parameter controller [142].

The impedance controller is typically more effective for lightweight, backdrivable external devices (such as cable-driven devices) compared to other controllers [124]. It extends the position control, enabling it to not only regulate the position and force but also the relationship and interaction between the exosuit and the human body [142,143]. This controller architecture includes an impedance module, which receives the error position of the joints and yields the force values that serve as force references for subsequent stages. The architecture also comprises a force/torque controller that attempts to ensure that the forces exerted by the exoskeleton actuators are approximately equal to force references [121].

The admittance controller is employed to regulate the force generated by the external skeleton during interaction with the user [144]. It features an admittance model, which receives forces and outputs the position, as well as a position controller that controls the joint angle based on position references from the admittance model output [121].

Wu et al. [145] implemented a physical parameter-based control system in an exoskeleton for upper-limb rehabilitation of disabled patients. They used a modified sliding mode control strategy incorporating a proportional integral derivative (PID) sliding surface and a fuzzy hitting control law to ensure a robust and optimal position control performance. Their approach led to the best control performances in terms of tracking accuracy, response speed, and robustness against external disturbances.

The **usage-based control systems**, such as those implemented in MGA [146] and L-Exos [147], can be categorized into three types: virtual reality (VR) controller, teleoperation controller [121], and gait controller, which is commonly used in lower limb solutions [140]. VR controllers are commonly employed in rehabilitation exercises for upper-limb exoskeletons [148]. They allow for the guidance and assistance of patients during tasks such as moving a virtual object with their hands [139], virtually painting a wall [146], or carrying out constrained motion tasks [147]. In these applications, the exoskeleton/exosuit can be regarded as a haptic device [121].

The teleoperation controller is a form of master-slave controller, where the exoskeleton worn by the user is commonly used as the master type and a mirror robot serves as the slave [121]. In this configuration, interaction control occurs between the slave robot and the environment, as opposed to the typical interaction between the user and the exoskeleton [121]. Rahman et al. [149] implemented a teleoperation controller in an exoskeleton for rehabilitation and passive arm movement assistance (MARSE-4), constituted by an upper-limb prototype and a master exoskeleton arm (mExoArm). While mExoArm is operated by the patient, the upper-limb prototype mirrors the movement.

Liu et al. [150] developed and implemented a novel systematic algorithm of gait control based on energy efficiency. Their ultimate goal was effectively to reduce the high energy consumption of devices.

#### 2.4.2. Sensors

Capturing human motion intents for external device control is a major challenge, which can be addressed through the use of sensors associated with both the control system and the device [151]. These sensors capture the user’s movement intention as an input signal to the control system, which then provides output to the exosuit to perform the intended move. To ensure success, this input signal must be precise and accurate. In addition to the intention-prediction instrumentation, other sensors such as inertial measurement units [152] (e.g., gyroscopes [153] and accelerometers [154]) or mechanical sensors [155] can be employed to measure or evaluate the output movement. However, it should be noted that these sensors are unable to predict movement beforehand [156].

Several control methods have been proposed to detect human intention through human–robot interaction dynamics, which could effectively assist able-bodied human subjects [157,158]. While control methods using human–robot interaction dynamics are effective in assisting able-bodied humans, they may not always be suitable as the user needs to produce sufficient torque at joints to initiate movement. If this amount of torque is not generated, the device may not be effectively controlled, resulting in a problematic aspect for elderly or severely disabled individuals [159]. The ideal solution for human–robot interaction entails the prediction of movement intention, instead of a reaction to a precursor movement. This approach can improve performance in scenarios where generating sufficient torque is not possible [122].

To predict human movement, electrophysiological signals from proteins, organs, or muscles can be captured through sensors measuring voltage changes or electric current [160,161]. **EMG sensors** (intramuscular [162], surface [163]) can measure small electrical signals [164] produced by muscle contraction and have been successfully used in exoskeleton control [165,166]. EMG-based methods can capture the user’s intention to control the device, even if the person cannot produce sufficient joint torques or execute a particular movement [122]. However, the signal measured by EMG sensors might be biased by various factors, such as muscle crosstalk susceptibility [151], skin condition (surface sensors), muscle fatigue [156], or the inaccessibility of deep muscle fibers [167].

In addition to the use of EMG sensors, there are other sensors that can be considered to be alternatives for measuring muscle electrical activity. One such alternative is **mechanomyography** (MMG) sensors which are less sensitive to skin conditions compared to EMG sensors [156,168]. These sensors measure the signal produced by muscles with respect to the gross lateral muscle movements which causes low-frequency vibration during contraction, lateral vibrations at the muscle’s resonant frequency, and volumes introduced by the changes in the muscles [169]. Despite the advantages, MMG sensors have some disadvantages, such as being affected by muscle fatigue as well [156]. **Sonomyography** (SMG) sets up another possibility to predict the user’s movement intention by measuring muscle thickness and tracking skeletal muscle deformation from superficial to deep tissue [170,171]. SMG sensors are also capable of classifying several motions and predicting joint kinetics during dynamic activities, such as those in the wrist [171,172]. However, muscle fatigue is still a common issue with SMG sensors [156]. Figure 8 synthesizes the way these three techniques work.

Finally, **Electroencephalogram** (EEG) sensors can capture the user’s intention without using sensors that measure the signal produced directly in muscles [156,174]. Instead, they measure the electrical activity in the brain. However, the signal captured by EEG sensors is not accurate enough and can only be used for classifying movements [156]. Table 3 summarizes the sensors’ advantages and disadvantages.

All the above-mentioned possibilities capture analog signals, which need to be further converted into digital signals before being sent as input to actuators. This conversion can be performed with affordable solutions such as an Arduino [175] and with commercial solutions already developed, such as BITalino from pluX [176,177] or TMSi products [178,179].

## 3. Device Solutions

There are currently various exoskeleton and exosuit solutions available, not only those described in the literature but also available commercially, such as Rewalk [180], Ekso [181], Cyberdyne [182], RB3D [183], and others. These solutions have been designed as powered or passive wearable devices that can assist individuals in daily living activities, including walking assistance. However, the present review will only focus on solutions discussed in the literature specifically related to ankle/foot and hand/arm examples. The structural materials used, actuation systems, control approaches, implementation, and results achieved will be described in detail.

### 3.1. Ankle/Foot Solutions

An external device known as an ankle-foot orthosis (AFO) is commonly prescribed to treat ankle impairments [184] while also helping to facilitate walking, which is essential to daily living routines. The use of an AFO has also been shown to reduce the metabolic cost of movement while rehabilitating weak ankles and feet [184]. Patients with ankle disabilities typically experience weakness in the muscles associated with plantar flexion and/or dorsiflexion movements, as illustrated in Figure 9. Debility in the gastrocnemius, soleus, and plantaris muscles, which are involved in plantar flexion movement, may reduce the push-off power necessary to propel the body forward from the stance phase [184]. Additionally, weakness in the tibialis anterior muscle, which is involved in dorsiflexion movement, may result in a drop-foot gait during the swing phase due to an inability to adequately lift the toes [184].

Various procedures, including surgical, therapeutic, and orthotic, can be used to treat ankle impairments. Recently, orthotic procedures have become the most commonly used [184]. In such cases, the device must be attached to the wearer and aligned with their ankle and foot to assist weak or paralyzed muscles by generating torques or forces [184]. Assistance can be provided using passive, semi-passive, or active AFO exosuits, depending on the availability of an energy source [184]. For example, Yamamoto et al. [185] and Ramsey et al. [186] have developed passive devices, while Furusho et al. [187], Mooney et al. [83], Takahashi et al. [188], and Dong et al. [189] have developed semi-passive or active ankle solutions. In addition to these authors, there are other examples of AFO devices described in the literature that provide a better understanding of what has been developed. Awad et al. [190] developed a lightweight (0.9 kg), powered, and soft wearable ankle exosuit that interfaces with the paretic limb. The exosuit is composed of functional textile anchors that are visually similar to normal clothes. The actuation method is achieved using contractile Bowden cables located in the posterior and anterior anatomic planes of the ankle joint, allowing for plantar flexion and dorsiflexion movements, respectively. A low-profile shoe insole mechanically transmits power during walking. The cables are tensioned or relaxed through a body-worn actuator and a battery attached to a waist belt. The solution proposed in this study is capable of reducing the energetic burden associated with walking movement in individuals post-stroke, which under normal conditions can cost over 60% more than usual.

The exosuit’s control system used a combination of position measurements from linear potentiometers and force measurements from load cells integrated into the textiles. This instrumentation was combined with rotational velocity measurements from a gyroscope mounted in each shoe to adapt the Bowden cable position trajectories and generate the desired assistive force profile on an iterative basis. The gyroscopes enabled real-time gait segmentation, while the potentiometers and load cells enabled iterative, force-based, and position control. Together, these sensors enabled appropriately timed assistive forces with adequate magnitude. However, the control system only provides reactive help and is not suitable for individuals with severe paralysis [191]. Etenzi et al. [191] developed a lightweight unpowered passive-elastic exoskeleton made of aluminum, weighing 1.4 kg, which stores elastic energy in springs (two for each leg) that assist during walking. The energy is stored from knee extension to the end of the leg swing phase and is then released during ankle plantar flexion. The actuation control uses a ratchet and pawl system to store and return energy through compression and release phases of metal springs, which act simultaneously with the knee and ankle. This approach achieved a reduction in metabolic cost, using 11% less energy compared to disengaging the springs. However, compared to walking without the exoskeleton, the metabolic cost increased by 23%. Galle et al. [192] developed and tested a bilateral external device weighing only 0.890 kg. It consisted of an AFO at each leg, with a hinge at the ankle, and actuated through pneumatic artificial muscles connected between the foot and shank segments. The actuators were contracted when inflated with compressed air and aided during plantar flexion movements, achieving a 12% reduction in metabolic consumption compared to walking without the external device. The exoskeleton’s control was realized by an iterative learning algorithm that used the signal from load cells connected between the orthoses and pneumatic muscles as input, and linear displacement sensors placed between the foot and shank sections of the exoskeleton.

Bougrinat et al. [193] developed a 2.045 kg ankle-powered exoskeleton that provides at least 30 Nm of assistive plantar flexion torque using an electrical motor and Bowden cables attached from the user’s waist to carbon fiber struts fixed on the boot. They implemented a hierarchical architecture control system in an off-board personal computer for controlling the device. The high-level microcontroller estimates the gait cycle percentage by dividing the time passed in each cycle by the average walking period measured over ten cycles. The force-sensitive resistors placed under the insole at the heel area provide the needed input signals. The microcontroller then communicates to the PC, which is also a high-level controller, to transmit the desired current profile to the motor driver/encoder, which is a low-level controller. This particular exoskeleton was able to reduce the metabolic cost associated with the soleus and gastrocnemius muscles by 37% and 44%, respectively, [193].

The previous examples of ankle/foot external devices are summarized in Table 4, which shows a clear trend toward developing lightweight solutions. Figure 10 illustrates a generic scheme of the solutions described.

### 3.2. Hand/Arm Solutions

Brown et al. [194] illustrates the initial use of hand external devices to aid people with paralysis. Subsequently, such devices were employed in rehabilitation environments [195], particularly for individuals diagnosed with neurological disorders [196]. According to Ferguson et al. [195], hand exoskeletons or exosuits can be classified into four categories: assistive, rehabilitation, augmentation, and virtual reality. Assistive hand exoskeletons, such as those developed by Lucas et al. [197] and In et al. [198], aim to reduce muscular fatigue and improve functional dexterity [199]. Due to their portable design, these devices typically have fewer and smaller actuators, resulting in a more lightweight solution.

Ferguson et al. [195] explains that rehabilitation hand devices, such as those developed by Wege et al. [200] and Kawasaki et al. [201], are not required to be portable, as they are typically intended for use in physical therapy by multiple individuals. However, this requirement and the need to accommodate multiple DOF, impairs the conception and development of these devices. Typically, as the complexity of the solution increases, so does the weight of the device.

Ferguson et al. [195] noted that augmentation exoskeletons, such as those developed by Shields et al. [202] and Hasegawa et al. [203], aim to improve the physical abilities of able-bodied individuals. However, designing such devices entails significant challenges, such as minimizing their weight while still reproducing the DOF of a healthy hand. Currently, there is no combination of mechanical structural materials and actuators or power supplies that can provide a meaningful augmentation force.

There is another category of hand exoskeletons [195] that differs from the other types, as their goal is not to assist or enhance hand movements. Instead, they aim to simulate interaction through VR handsets by using haptic devices [204]. Park et al. [205] prototyped a dual cable hand exoskeleton to serve as an interface for VR environments. The device just weighs 320 g and can feedback on the touch sensation of hard and soft objects.

Yap et al. [206] developed a soft robotic assistive glove for individuals with grasp pathologies to assist them with everyday activities. The device is capable of supporting various hand manipulation tasks, including finger and thumb movements during hand closing and grasping activities. The glove is actuated by low-profile, soft, elastomeric pneumatic actuators that require low pressure.

The manipulation control approach involves an EMG strategy associated with radio-frequency identification (RFID) to predict the user’s intentions. RFID tags act as non-physical switches that enable the activation of different hand gestures. Subsequently, the Arduino microcontroller receives the input from the sensors, and the voltage regulator sends output to the pressure sensors and miniature pneumatic pumps for air pressure regulation.

Díez et al. [207,208] developed a modular hand exoskeleton for a rehabilitation environment that was originally designed for VR environments but was later adapted for real-life scenarios. The device is made using polylactic acid (PLA) 3D printable material and actuated by electric linear actuators placed in each finger. The exoskeleton is governed by a high-level controller that relies on EMG input signals. This control approach performs successfully in 97% of the trials [207], effectively triggering the opening and closing gestures.

Agarwal et al. [128,209] developed a unique solution that differs from previous studies by considering three closed-loop chains to manipulate the four DOF of the thumb. Specifically, the DOF comprise carpometacarpal (wrist) flexion-extension and abduction-adduction movements, MCP flexion-extension movements, and interphalangeal flexion-extension movements (shown in Figure 1 and Figure 2). This closed-loop approach also resolves issues of axis misalignment at the exoskeleton-human joints.

The actuation method of Agarwal et al. [209] employs Bowden cables connected to actuated joints, enabling the transfer of up to 0.4 Nm of torque to each exoskeleton joint, producing highly backdrivable actuators with low reflected inertia and a weight of approximately 30 g each.

Each exoskeleton joint is equipped with a pulley that has a cable attached to its circumference. The cables are pulled by a brushed DC motor, which regulates the torque of each exoskeleton joint through a PID controller. This controller tracks the desired value, ensuring that each thumb joint and movement has a root mean square error of no greater than 13%. Structurally, the device was produced using selective laser sintering, which made it lighter, with some metallic parts added for load-bearing strength and durability.

According to Agarwal et al.’s findings [128,209], the device aligns with the natural movements of all thumb joints.

The following example also involves the development of a glove exosuit for hand rehabilitation by Klug et al. [210]. The device uses structural materials, such as microfibers, elastics, and PU pleather. It weighs 0.435 kg including batteries and controllers. Wires located along the palmar and dorsal sides of the hands, resembling flexor and extensor tendons, respectively, actuate the glove, allowing independent finger movements. These wires are pulled by an electrical DC servomotor capable of transmitting up to 20 N of force. The exosuit is controlled through the readings of force sensors placed at the fingertips. In some situations, this approach may limit comfort and touch sensitivity while it provides a rough force estimate. The solution depends on two distinct sensor technologies, one based on piezo-resistive bending elements mounted dorsally, and the other on electroactive-based polymers located dorsally and on the palm. Using machine learning algorithms fed by the sensor readings, the exosuit controller regulates force almost in real time. Consequently, the hand exosuit is capable of producing a maximum force of 27.4 N, assuming both the user and the device forces, and a mean bending angle of 132°.

Table 5 provides a concise summary of the hand/arm exosuit examples considered. An illustrative generic representation of an exoglove is shown in Figure 10.

## 4. Ethical Issues

The adoption of external devices, such as exoskeletons or exosuits, for the purpose of enhancing physical abilities, whether for military, industrial, or rehabilitation contexts, could potentially introduce ethical, social, and legal issues to individuals and society [211,212]. While there are undoubtedly numerous benefits associated with the use of these devices, including the ability to provide individuals with physical impairments greater freedom of movement and to increase safety conditions in the workplace, it is crucial to ensure proper regulation and monitoring to mitigate any potential negative consequences.

There is a worldwide tendency from both military forces and industrial companies to increasingly adopt the use of external devices to augment the physical capabilities of their soldiers and employees. However, there is a growing concern that these devices may result in the dehumanization of their users, as their primary aim became to achieve greater efficiency, endurance, and productivity in combat and work contexts, respectively, [213].

Currently, the high cost and experimental nature of using external devices for mass rehabilitation purposes renders them inaccessible to the majority of the global population [213]. This scenario brings to light ethical concerns related to the potential of this technology to amplify existing social inequalities, and for being the source of new ones—such concerning prospects deserve an integrated societal response by all the relevant stakeholders.

## 5. Present and Future Perspectives

According to Bao et al. [214], the overall number of scientific publications about robotic exoskeletons increased exponentially since the 1990s. Among the research areas (orthopedics, computer science, automation, and control) pushing this topic forward, engineering/biomedical and rehabilitation fields take the lead. Therefore, it is reasonable to expect an evolution of exoskeletons and exosuits in the near future. The review study published by Hill et al. [215] points to the potential of the technology employed in such devices, to improve the functional capabilities of individuals with neurological impairment, particularly in relation to ambulatory outcomes.

Despite their potential for mass adoption, the majority of devices found in the literature are still prototypes or academic examples. Zhang et al. [216] reviewed and compared several lower limb orthoses for the rehabilitation of patients with SC injuries. From the analysis, only one was evaluated with an A grade for recommendation. A similar investigation was made by Miguel-Fernández et al. [217]. They evaluated the control strategies used in exoskeletons for gait rehabilitation in more than a thousand scientific papers. In the end, they noticed a low effectiveness of those control systems on clinical outcomes, justified by a lack of standardization in the experimental protocols which leads to high levels of heterogeneity. We see this heterogeneity as a consequence of the exploratory nature of the research in this domain. After this preliminary research stage, as resources, such as artificial intelligence get a deeper integration into the control/actuation processes and help mitigate the current shortcomings of existing technologies, a second generation of these devices is expected to emerge. In this phase, standards and regulations both for testing and usage, are expected to emerge. The current paper aims to provide a valuable reference tool, instrumental in facilitating this evolution. Another avenue for improvement, according to Oña et al. [218], depends on a better symbiosis with VR technologies required to promote a long-term recovery of motor function in daily living activities. Moreover, the recent pandemic situation caused by COVID-19 stressed the need for continuous and reliable rehabilitation therapeutics, pointing to home-based recovery solutions [219]. During home rehabilitation time, these devices might need to be worn while performing current daily tasks. As pointed out by Wolff et al. [220], citing stakeholders such as healthcare professionals, such devices need to allow toileting, getting in and out of the car, climbing stairs, etc.

The recent advances in artificial intelligence and machine learning have also improved mobile robotic exodevices used in motor rehabilitation. According to Vélez-Guerrero et al. [221], there is a latent need to develop more reliable systems through clinical validation and improvement of technical characteristics.

Despite the long journey that rehabilitation devices have already taken, such as reducing hospital costs and improving the overall well-being of their users, there are still flaws and gaps that must be solved to address current and future needs.

## 6. Conclusions

A concise review of the state of the art in exoskeleton and exosuit rehabilitation solutions was presented. The materials used in the structure, the actuators (and their associated power sources), and the control systems, as described in this review, can be combined in various configurations to fabricate external devices aimed at enhancing or rehabilitating the physical capabilities of humans.

The exoskeleton and the exosuit are different concepts that can potentially be combined to address their respective limitations. For instance, leveraging the advantages of soft structural materials with the enhanced performance of traditional actuation methods can produce optimal results.

Nevertheless, it is fundamental to acknowledge that external skeletons and suits have distinct objectives dictating their design. Taking lower limb devices as an example, exoskeletons are primarily intended to bear an individual’s weight, as in the case of lower limb paralysis, whereas exosuits are typically only capable of assisting with movement if there is some mobility (tough residual) in the legs.

The authors of this review are confident that they have exhaustively explored the most relevant examples pertaining to the topics discussed. With this effort, they hope to have contributed in some measure toward promoting a faster and more effective development of external devices for the benefit of humanity.

## Figures and Tables

**Figure 1 bioengineering-10-01328-f001:**
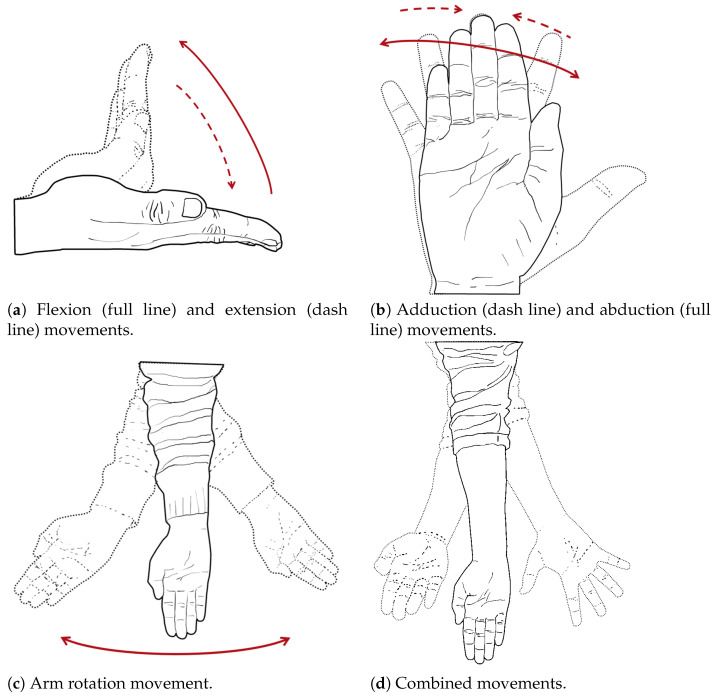
Basic human hand movements [54] and their combination toward complex movement.

**Figure 2 bioengineering-10-01328-f002:**
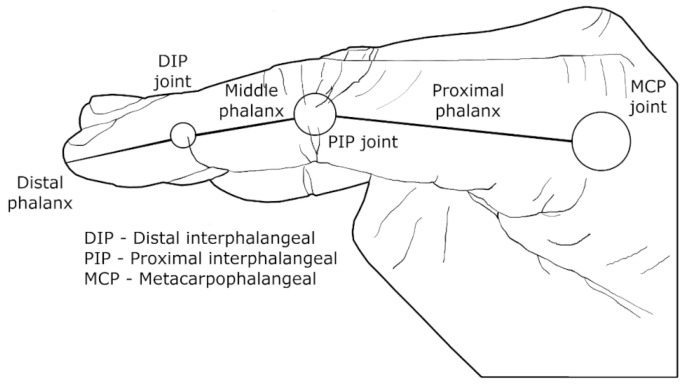
Finger joints and phalanges—the 4 DOF of a finger wearable [60].

**Figure 3 bioengineering-10-01328-f003:**
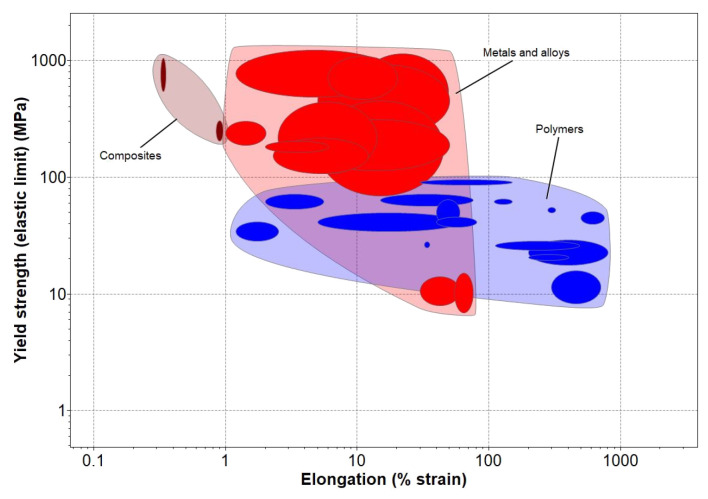
Yield strength (elastic limit) vs. Elongation of traditional and soft materials, Granta Edupack 2020 [71].

**Figure 4 bioengineering-10-01328-f004:**
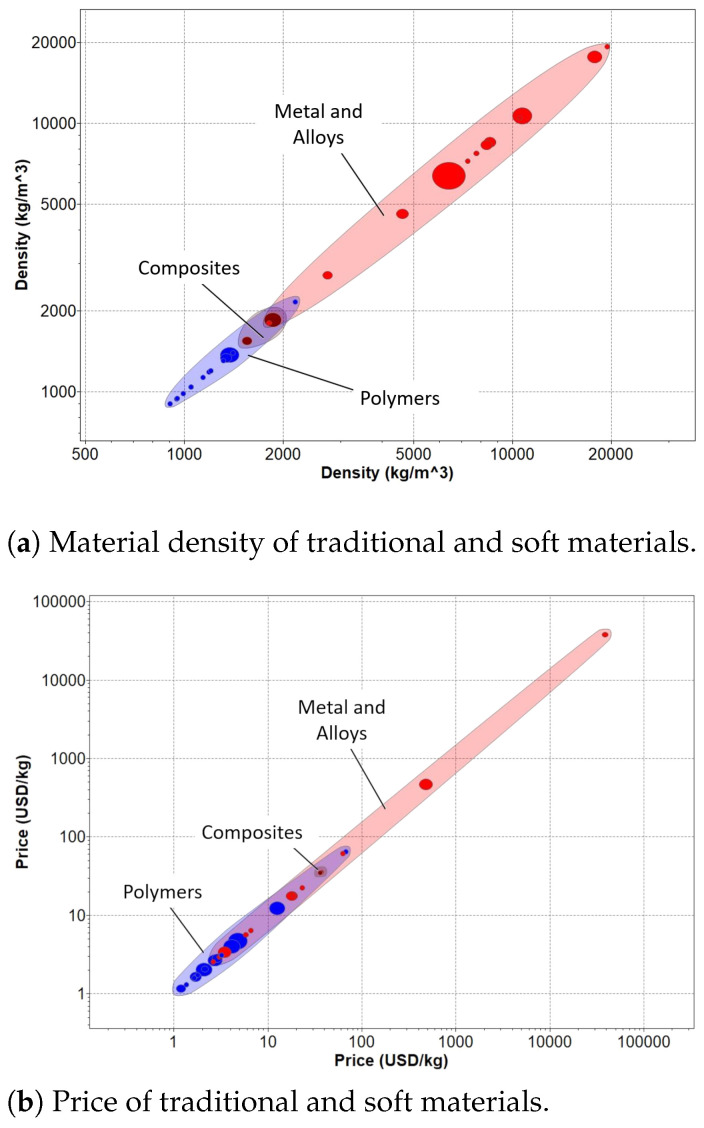
Important considerations regarding traditional vs. soft materials, Granta EduPack 2020 [71].

**Figure 5 bioengineering-10-01328-f005:**
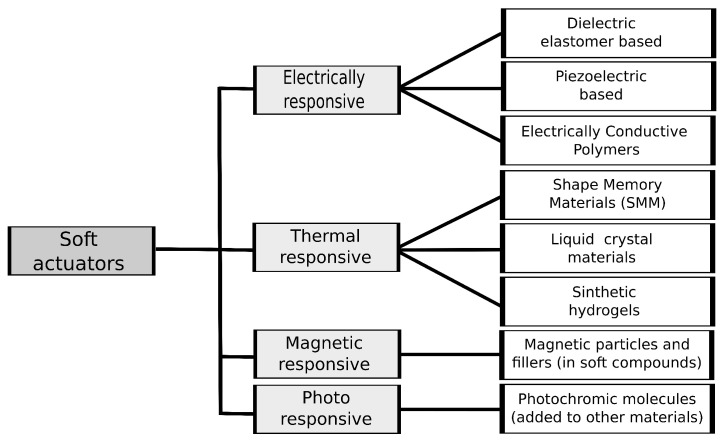
Soft actuators, responsiveness to stimuli, and base materials.

**Figure 6 bioengineering-10-01328-f006:**
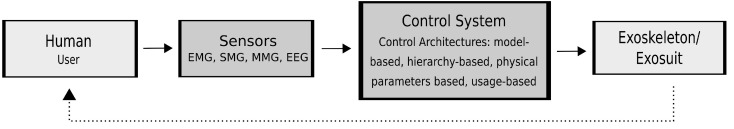
Control system overview.

**Figure 7 bioengineering-10-01328-f007:**
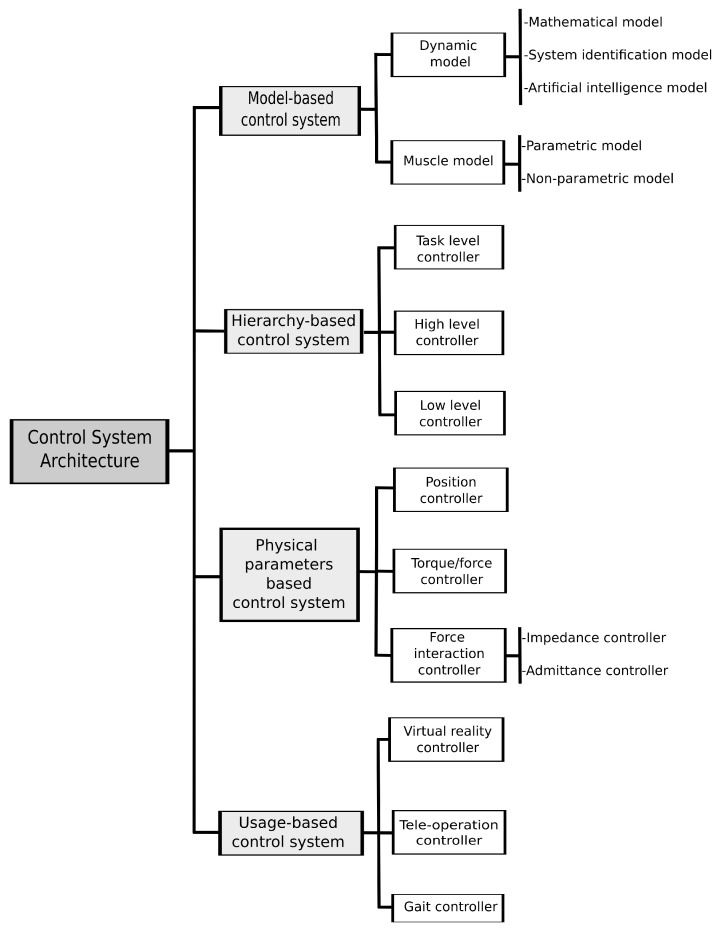
Typical control system architectures [121].

**Figure 8 bioengineering-10-01328-f008:**
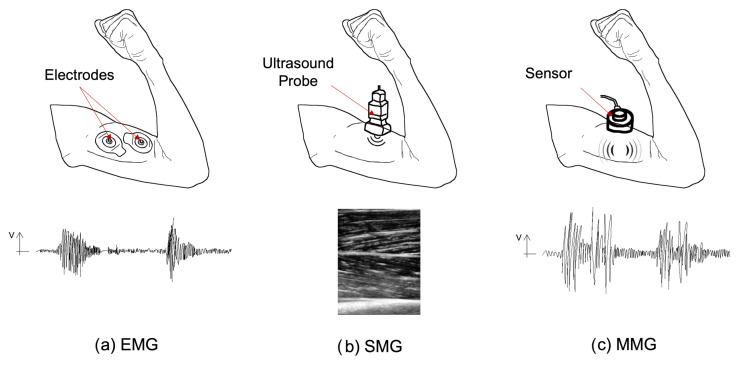
Sensors on muscles and respective outputs (SMG output [173]).

**Figure 9 bioengineering-10-01328-f009:**
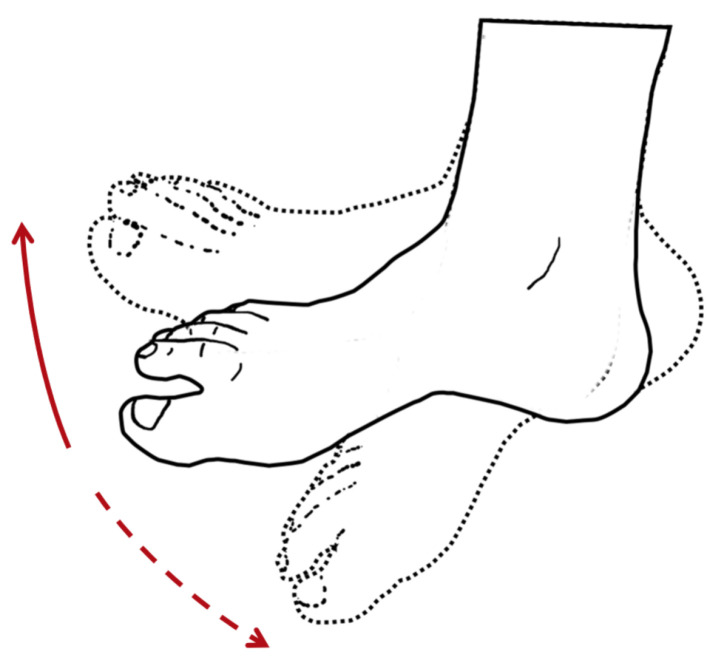
Dorsiflexion (full line) and plantar flexion (dash line) movements.

**Figure 10 bioengineering-10-01328-f010:**
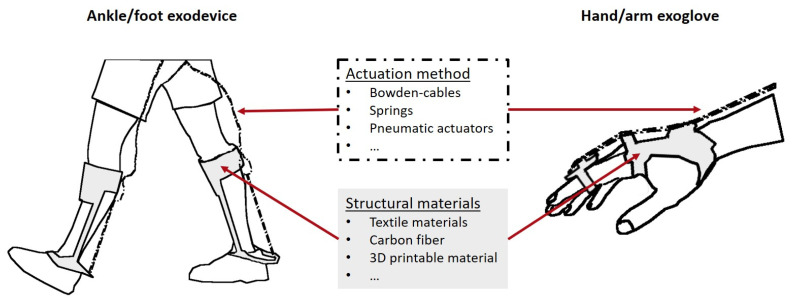
Generic illustration of an ankle/foot exodevice and a hand/arm exoglove (black dash-dot line represents the actuation method).

**Table 1 bioengineering-10-01328-t001:** Advantages and disadvantages of Rigid vs. Soft materials.

Type of Materials		Advantages	Disadvantages	References
			Higher weight	
	Aluminium	Higher mechanical strength	Diminished ergonomy and comfort	
Rigid	Stainless steel	Higher elastic limit	Larger inertias	[65,69,70,71,72,73,74,75]
	Titanium	Higher safety	Unnatural motion patterns	
			Lead to higher power consumption	
		Safer		
		Allow smoother movements		
		Higher comfort		
	Polymers	Higher portability and flexibility	Lower yield strength	
Soft	Composites	Lightweight	Actuators with lower force/torque and velocity	[57,66,68,69,73,76,77,78,79,80]
	(e.g., SMPs, EAPs)	Biomimetic	Adequate for smaller assistance levels	
		Accommodate large deformations		
		Possible use as actuators		
		Easy to process and mass produce		

**Table 2 bioengineering-10-01328-t002:** Advantages and disadvantages of traditional and soft actuators.

Type of Actuators		Energy Source			Advantages	References
Traditional actuators	Purely mechanical actuators	Unpowered			No need for an external source of energy Allow reducing metabolic consumption	[44,84,86,87]
	Mechanical servomotor-based actuators	Powered—electrical input			High-efficiency power conversion Quiet, clean, and create no pollution Less expensive and easy to maintain Easy to implement the remote-control system No limitation of separation between the energy source and system	[88,89]
	Pneumatic actuators	Powered—compressed gas			Affordable Fast working cycle Insensitive to temperature drift No need for mechanical transmission High actuating forces	[89,90,91]
	Hydraulic actuators	Powered—compressed fluid			High stability High stroking velocity Suitable for high loads High actuating force Stiff and incompressible source	[92,93]
Soft actuators	Electrical responsive actuators	Powered—electrical stimulus	Dielectric actuator		Soft, flexible, and stretchable Scalable High power-to-weight ratio Stores and recovers kinetic energy	[89,94,95,96,97,98]
			Piezoelectric actuator		Suitable for high force applications Large operation bandwidth	[77,99,100,101,102,103,104]
			Conducting polymers		Possibility of being fed through biofuels Processability Good biological muscles emulation	[105,106,107,108]
	Magnetic responsive actuators	Powered—magnetic stimulus			Linear effect Quick response Capacity to penetrate most materials	[89,109,110,111]
	Thermal responsive actuators	Powered—thermal stimulus	SMM	SMPs	Low cost Biodegradable Low density High elastic deformable Sustain a broad range of temperature drift	[89,112,113]
				SMAs	Flexible in nature High energy density Low actuation temperature Provides large frequency response	[114,115,116]
	Photo-responsive actuators	Powered—light stimulus			Environmentally friendly Full possibility of remote control Easy to control the response Excellent resolution	[89,117,118]

**Table 3 bioengineering-10-01328-t003:** Sensors, their advantages and disadvantages.

Sensors		Advantages	Disadvantages	References
EMG	Measures the electrical signals from the muscle contraction	Predict movement intension even if with any movement performed Already tested	Biasable by muscle crosstalk susceptibility, skin conditions, muscle fatigue	[122,151,156,162,163,164,165,166,167]
MMG	Measures vibration and volume by changes in muscles	Less sensitive to skin conditions	Biasable by muscle fatigue	[156,168,169]
SMG	Measures thickness and deformation of muscles	Able to classify several motions and predict joint kinetics during dynamic activities	Biasable by muscle fatigue	[156,170,171,172]
EEG	Measures electrical activity in the brain	No need for sensors in the muscles	Not enough accuracy	[156,174]

**Table 4 bioengineering-10-01328-t004:** Examples available in the literature for ankle/foot solutions.

	Weight	Structural Materials	Actuation Method	Control System	Results	References
Awad et al.	0.9 kg	Textile materials	Powered—Bowden cables	IMU and load cells	Reduces the metabolic cost	[190]
Etenzi et al.	1.4 kg	Aluminium	Unpowered—Springs	Mechanic	Increases the metabolic cost in 23%	[191]
Galle et al.	0.89 kg	-	Powered—Pneumatic actuators	Iterative Learning Algorithm, load cells and IMU sensors	Reduces the metabolic cost in 12%	[192]
Bougrinat et al.	2.045 kg (considering all components)	Carbon fiber	Powered—Bowden cables	Hierarchic Control Architecture	Reduces significantly the metabolic cost of the plantar flexion muscles	[193]

**Table 5 bioengineering-10-01328-t005:** Hand/arm exosuit applications found in the literature.

	Type	Structural Materials	Actuation Method	Control System	Results	References
Yap et al.	Assistive	Elastomers textile gloves	Pneumatic Actuators	EMG RFID	Satisfactory results Maximum force achieved 1.57 N	[206]
Díez et al.	Rehabilitation	3D printable material PLA	Electric linear Actuators	EMG controller	97% success during the trials	[207,208]
Agarwal et al.	Rehabilitation	Selective laser sintering materials Metallic load bearing parts	Bowden cables with springs Brushed DC motor	-	Compatible natural motion solution Max. torque 0.4 Nm	[128,209]
Klug et al.	Rehabilitation	Glove—microfibers, elastics and PU pleather	Wires Electrical motor	Force sensorsMachine learning algorithm	Max. angle motion 132° Max. force 27.4 N	[210]

## Data Availability

Not applicable.

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
