# Peer review of "Exo Supportive Devices: Summary of Technical Aspects"

_bioengineering, 2023, doi:10.3390/bioengineering10111328_

Round 1

Reviewer 1 Report

Comments and Suggestions for Authors - My main concern is on the scientific value and usefulness of the review because the real essence of Challenges and Potential Opportunities is very limited. Section 5 is less than half a page! Please extend this section and make it critical and comprehensive such that it conveys new research directions to potential readers. - The scope of the research needs to be explicitly mentioned. e.g. does it cover upper extremity support devices or lower extremity or both? It seems to be addressing both. The paper seems to be covering both domains and is a bit generic. I would like to know the opinion of the authors on this comment. - The discussion on the pivotal role of exoskeletons can benefit from state-of-the-art works "A human hand compatible optimised exoskeleton system" and 'Stroke rehabilitation using exoskeleton-based robotic exercisers: Mini Review.' - There is a lot of bookish knowledge which needs to be reduced e.g. intro to rigid and soft materials, advantages of both etc. - The document structure needs to be improved especially if there are several small paragraphs that need to be merged to enhance the continuity. - With the website references, please update the accessed date.  - Arrange list of abbreviations (at the end of the paper) in alphabetical order. Comments on the Quality of English Language

Moderate English language changes are required.

Reviewer 2 Report

Comments and Suggestions for Authors

1. Please provide the appropriate literature citations for the description of Figure 2 on page 5;

2. To enhance the comprehensiveness of Table 1, include references from relevant literature that have employed the corresponding materials. Similarly, please ensure proper citations in Table 2 to indicate which devices that have previously utilized that specific actuator;

3. Adjust the sequence of some figures, including Figure 9, which was mentioned before Figure 5 through 8;

4. The control system depicted in Figure 6 appears too generalized. please provide additional details;

5. Additionally, include a reference in the description of Figure 7;

6. If possible, consider adding a table to compare the features of different sensors discussed in the article;

7. As with Tables 1 and 2, please ensure proper citations to enhance the readability of Tables 3 and 4. 

Comments on the Quality of English Language

1. Please modify the description of Figure 4(a), which is hidden behind Figure 4(b). The description of Figure 4(b), "Price of of traditional and soft materials.", also contains grammar and typographical problem;

2. In the reference section, there are frequent issues of incompleteness or improper usage of reference, including, but not limited to [3], [9], [37], [43],[52], [62], [81], [97], [100], [103], [105], [106], [113], [127], [131], [134], [135], [138], [140], [147], [159], [167], [180], [182], [196], [199], [203], and [210]. 

Reviewer 3 Report

Comments and Suggestions for Authors

Dear Authors,

I have reviewed the following manuscript: Exo Supportive Devices: Summary of Technical Aspects.

This paper - as a review article - deals with devices for the external movement of human limbs. The authors present these devices and their elements in detail, for which the authors use 211 literature. The article presents and compares the types of materials that can be used, the traditional and innovative actuators and control options. The chapter dealing with control seems superficial, the application of specific types of control (e.g. sliding mode control, fuzzy control, etc.) would improve the quality of the article. In addition to actuators, the article also deals with sensors.

The only strong shortcoming of the article is the visual presentation of the devices, the visual appearance of some specific devices would break the monotony caused by a lot of text.

To sum up, the article is logically structured, the chapters are well built on each other, and indicates thorough research work. I will consider the corrections mentioned above to improve the article.

Round 2

Reviewer 1 Report

Comments and Suggestions for Authors

Authors have addressed the comments suggested earlier.

Comments on the Quality of English Language

Moderate editing of English language required.

Reviewer 2 Report

Comments and Suggestions for Authors

The reviewer is now satisfied with the current version of the article.